# Soluble C-Type Lectin-Like Receptor 2 Is a Biomarker for Disseminated Intravascular Coagulation

**DOI:** 10.3390/jcm10132860

**Published:** 2021-06-28

**Authors:** Akitaka Yamamoto, Hideo Wada, Yuhuko Ichkawa, Motoko Tanaka, Haruhiko Tashiro, Katsuya Shiraki, Hideto Shimpo, Yoshiki Yamashita, Takeshi Mastumoto, Motomu Shimaoka, Toshiaki Iba, Katsue Suzuki-Inoue

**Affiliations:** 1Department of Emergency and Critical Care Center, Mie Prefectural General Medical Center, Yokkaichi 510-0885, Japan; akitaka-yamamoto@mie-gmc.jp (A.Y.); haruhiko-tashiro@mie-gmc.jp (H.T.); 2Department of Central Laboratory, Mie Prefectural General Medical Center, Yokkaichi 510-0885, Japan; ichi911239@yahoo.co.jp (Y.I.); motoko@josai.ac.jp (M.T.); katsuya-shiraki@mie-gmc.jp (K.S.); 3Department of General Medicine, Mie Prefectural General Medical Center, Yokkaichi 510-0885, Japan; 4Mie Prefectural General Medical Center, Yokkaichi 510-0885, Japan; hideto-shimpo@mie-gmc.jp; 5Department of Hematology and Oncology, Mie University Hospital, Mie University Graduate School of Medicine, Tsu 514-8507, Japan; yamayamafan4989@yahoo.co.jp; 6Department of Blood Transfusion and Cell Therapy, Mie University Hospital, Mie University Graduate School of Medicine, Tsu 514-8507, Japan; matsutak@clin.medic.mie-u.ac.jp; 7Department of Molecular Pathobiology and Cell Adhesion Biology, Mie University Graduate School of Medicine, Tsu 514-8507, Japan; motomushimaoka@gmail.com; 8Department of Emergency and Disaster Medicine, Juntendo University Graduate School of Medicine, Tokyo 113-8421, Japan; toshiiba@juntendo.ac.jp; 9Department of Clinical and Laboratory Medicine, Yamanashi Medical University, Yamanashi 400-8510, Japan; katsuei@yamanashi.ac.jp

**Keywords:** platelet activation, sCLEC-2, DIC, poor outcome

## Abstract

Disseminated intravascular coagulation (DIC) is induced by excess activation coagulation, and activated platelets are also involved in pathogenesis. Therefore, plasma levels of soluble C-type lectin-like receptor 2 (sCLEC-2), a new marker for platelet activation, can be expected as a marker of DIC in critically ill patients. Plasma levels of sCLEC-2 and D-dimer were measured using the STACIA system. Plasma sCLEC-2 and D-dimer levels were significantly higher in patients with underlying diseases of DIC than in those with unidentified clinical syndrome (UCS). Plasma sCLEC-2 levels were significantly higher in the patients with DIC and Pre-DIC than in those without DIC or Pre-DIC. Similarly, plasma D-dimer levels were also significantly higher in patients with DIC and Pre-DIC than in those without DIC or Pre-DIC. The plasma sCLEC-2 levels in all patients and those with a DIC score ≤ 4 were significantly higher in non-survivors than survivors. The plasma D-dimer levels in all patients, those with a DIC score ≥ 5 and those with a DIC score ≤ 4, were significantly higher in non-survivors than in survivors. The plasma sCLEC-2 is expected as a marker for DIC/Pre-DIC as well as the prognostic marker in critically ill patients.

## 1. Introduction

Disseminated intravascular coagulation (DIC) is a fatal disorder commonly associated with infectious diseases, hematological malignancy, solid cancer, aortic aneurysm, trauma and obstetric diseases [1,2,3,4]. DIC causes disseminated microthrombi in the microvasculature, and severe DIC results in organ failure [5]. Laboratory findings in DIC indicate hypercoagulation/fibrinolysis, represented by elevated fibrinogen and fibrin degradation product (FDP) and D-dimer [6], and consumptive coagulopathy, with decreased platelet counts and a prolonged prothrombin time (PT) [7]. DIC is usually diagnosed by the scoring system based on the FDP/D-dimer value, PT, platelet count, and fibrinogen level [8].

As thrombotic microangiopathy (TMA) also causes disseminated microthrombi in the microvasculature and involves severe thrombocytopenia, organ failure and hemolytic anemia [9], the differentiation between TMA and DIC has been discussed [10]. Roughly speaking, DIC is caused by the blood coagulation system, while TMA is caused by platelet activation.

Although the β-thromboglobulin (β-TG), platelet factor 4 (PF4), and P-selectine are the candidates of biomarkers for platelet activation, their actual diagnostic specificity for thrombosis due to platelet activation is not high, and they have little clinical laboratory use. Soluble platelet membrane glycoprotein VI (sGPVI) and soluble C-type lectin-like receptor 2 (sCLEC-2) have thus been introduced as new biomarkers of platelet activation [11,12,13,14], and the elevation of sGPVI and sCLEC-2 is reported in TMA [11,14]. Other than TMA, elevated plasma levels of sGPVI have been reported in postoperative patients [12] and patients with acute coronary syndrome [15]. On the other hand, elevated sCLEC-2 levels have been reported in patients with acute coronary syndrome [16,17].

In the present study, plasma sCLEC-2 and D-dimer levels were measured in 575 patients with DIC/Pre-DIC or non-DIC but with the same underlying diseases to examine their diagnostic value in DIC.

## 2. Materials and Methods

The plasma sCLEC-2 and D-dimer levels were continuously measured in 575 critically ill patients with underlying diseases of DIC at Mie Prefectural General Medical Center from 1 August 2019 to 30 April 2020. There were 246 patients with infection, 75 with aortic aneurysm, 62 with trauma (including burn or hypothermia), 32 with cardiopulmonary arrest (CPA), 32 with solid cancer, 37 with obstetric diseases, 11 with DIC or Pre-DIC due to other underlying diseases (other DIC/Pre-DIC) and 80 with unidentified clinical syndrome (UCS). DIC was diagnosed using the diagnostic criteria for DIC established by the Japanese Ministry of Health, Labor and Welfare [18]. Patients with a DIC score 5 or 6 points were considered to have Pre-DIC. Other thrombic diseases, including stroke, acute coronary syndrome and venous thromboembolism, were excluded.

The study protocol (O-0057) was approved by the Human Ethics Review committees of Mie Prefectural General Medical Center, and informed consent was obtained from each patient. Blood sampling was performed on the day of admission for hospitalized patients or on the day of arrival for outpatients. In DIC or pre-DIC patients, blood samples were obtained at the onset and before treatment of DIC.

### 2.1. Measurement of sCLEC-2 and D-Dimer Levels

We measured the plasma sCLEC-2 levels via chemiluminescent enzyme immunoassay (CLEIA) using previously described monoclonal antibodies and the STACIA CLEIA system (LSI Medience, Tokyo, Japan). In brief, magnetic particles were coated with the anti-CLEC-2 monoclonal antibody 11D5. The plasma samples were then incubated with antibody-coated magnetic particles, and after being washed, they were incubated further with the alkaline-phosphatase-labeled anti-CLEC-2 monoclonal antibody 11E6. After being washed again, the magnetic particles were incubated with chemiluminescent substrate solution (CDP-Star; Applied BioSystems) and the luminescence was measured using the luminometer installed in the STACIA system.

The D-dimer levels were determined via the latex agglutination method using LPIA-ACE D-Dimer II (LSI Medience) [19]. The fibrinogen levels and PT-international normalize ratio (INR) were measured using an automatic coagulation analyzer (CS-5100, Sysmex, Kobe, Japan) using Thrombocheck Fib (L) and Thromborel S (Siemens Healthcare Diagnostics Products GmbH, Malvern, PA, USA).

### 2.2. Statistical Analysises

The data are expressed as the median (25th to 75th percentile). Differences between groups were examined for significance using the Mann-Whitney U test. *p*-values of ≤0.05 were considered to indicate statistical significance. The correlation among the sCLEC-2, D-dimer and platelet counts was examined using Spearman’s rank correlation coefficient. All statistical analyses were performed using the Stat flex software program (version 6: Artec Co. Ltd., Osaka, Japan).

## 3. Results

The plasma sCLEC-2 (mean ± standard deviation) level was 59.1 ± 16.7 pg/mL in 79 healthy volunteers. The plasma sCLEC-2 and D-dimer levels were significantly higher (*p* < 0.001) in patients with infection (379 pg/mL; 275–798 pg/mL and 4.7 μg/mL; 1.8–11.1 μg/mL, respectively), those with aortic aneurysm (307 pg/mL; 246–489 pg/mL and 7.4 μg/mL; 3.2–13.2 μg/mL), those with trauma (including burn or hypothermia) (247 pg/mL; 177–353 pg/mL and 6.2 μg/mL; 2.5–22.2 μg/mL, respectively), those with CPA (652 pg/mL; 440–1162 pg/mL and 23.6 μg/mL; 9.0–44.7 μg/mL, respectively), those with solid cancer (279 pg/mL; 194–365 pg/mL and 4.1 μg/mL; 2.5–15.1 μg/mL, respectively), those with obstetric disease (153 pg/mL; 207–276 pg/mL and 2.5 μg/mL; 1.7–7.8 μg/mL, respectively) and those with DIC or Pre-DIC due to other underlying diseases (424 pg/mL; 257–464 pg/mL and 21.6 μg/mL; 19.2–39.2 μg/mL, respectively) than in those with UCS (194 pg/mL; 146–250 pg/mL and 0.7 μg/mL; 0.4–1.6 μg/mL, respectively). The frequency of a DIC score ≥ 5 points was highest in patients with CPA and lowest in patients with obstetric disease. Mortality was high in patients with CPA and those with other DIC/Pre-DIC and was 0% in patients with obstetric disease and those with UCS (Table 1).

The plasma sCLEC-2 levels were significantly higher (*p* < 0.001) in patients with DIC (356 pg/mL; 234–500 pg/mL) and those with Pre-DIC (305 pg/mL; 226–441 pg/mL) than in those without DIC or Pre-DIC (246 pg/mL; 181–322 pg/mL) (Figure 1). The plasma D-dimer levels were significantly higher (*p* < 0.001) in the patients with DIC (28.5 μg/mL; 19.3–47.6 μg/mL) and those with Pre-DIC (18.5 μg/mL; 9.6–28.9 μg/mL) than in those without DIC or Pre-DIC (2.9 μg/mL; 1.1–6.9 μg/mL) (Figure 2).

The plasma sCLEC-2 levels in all patients and patients with a DIC score ≤4 were significantly higher in non-survivors (350 pg/mL; 238–476 pg and 319 pg/mL; 206–444 pg/mL, respectively) than in survivors (247 pg/mL; 181–328 pg and 244 pg; 180–319 pg/mL) (Figure 3). The plasma sCLEC-2 levels did not differ between DIC or pre-DIC patients with a ventilator (310 pg/mL; 240–468 pg/mL) and patients without ventilation (293 pg/mL; 193–468 pg/mL), or between patients with ≤28 days of hospitalization (321 pg/mL; 219–468 pg/mL) and those with ≥29 days of hospitalization (304 pg/mL; 227–475 pg/mL).

The plasma D-dimer levels in all patients, those with DIC scores ≥ 5 and those with DIC scores ≤ 4 were significantly higher in non-survivors (20.1 μg/mL; 9.5–41.4 μg/mL, 30.1 μg/mL; 17.8–47.1 μg/mL, and 3.8 μg/mL; 2.8–9.8 μg/mL, respectively) than in survivors (3.5 μg/mL; 1.3–8.9 μg/mL, 19.7 μg/mL; 10.1–28.6 μg/mL, and 2.7 μg/mL; 1.1–6.6 μg/mL, respectively) (Figure 4). The correlation of sCLEC-2 levels with the platelet counts was rS 0.170 (*p* < 0.001) and that of sCLEC-2 levels with D-dimer levels was rS 0.267 (*p* < 0.001).

## 4. Discussion

Disseminated microthrombi formation in the microvasculature is recognized in DIC. Although DIC is generally considered to be induced by the activation of coagulation [20], the activation of platelets may also be responsible for DIC. sCLEC2 levels are essential platelet-activating receptors in thrombosis, and elevated levels have been reported in many cases for TMA and a small number of cases for DIC [11,14].

The limitation of this retrospective study is that the measurement of sCLEC-2 was performed in patients with heterogenous underlying conditions. Various treatments may affect a patient’s outcome. However, in the present study, sCLEC2 and D-dimer levels were significantly higher in patients with underlying diseases of DIC, including DIC or Pre-DIC, than in patients with UCS. These findings suggest that both the coagulation system and platelets are activated by the underlying diseases of DIC, such as infection, CPA, trauma, aortic aneurysm, obstetric disease, and solid cancer. Under such conditions, antiplatelet therapy may be useful for the prevention of thrombosis [21].

Both the plasma sCLEC2 and D-dimer levels were significantly higher in patients with DIC or Pre-DIC than in those without DIC, suggesting that both the coagulation system and platelets are also activated in the onset of DIC. The plasma D-dimer levels were significantly higher in patients with DIC than in those with Pre-DIC, but the plasma sCLEC-2 levels showed no such pattern. The sCLEC-2 level may be useful for diagnosing the early state of DIC (as Pre-DIC). Although plasma D-dimer levels may relate to the progression of DIC, the relationship of the sCLEC-2 levels with the progression of DIC is unclear. These data suggest that anticoagulant therapy may be useful for treating DIC [1], but antiplatelet therapy is not.

Regarding the outcomes of all cases and patients with DIC or Pre-DIC, both the sCLEC-2 and D-dimer levels were significantly higher in non-survivors than in survivors. As these markers were high in DIC or Pre-DIC with a poor outcome [20], complication with DIC may itself be a cause of a poor outcome. The strong activation of coagulation and platelets may carry a high risk of a poor outcome in DIC or pre-DIC. Among patients without DIC or Pre-DIC, D-dimer levels were higher in non-survivors than in survivors, indicating that thrombosis exception with DIC may carry a high risk of a poor outcome. As this study population was heterogenous, there were no significant differences in the plasma sCLEC-2 levels between patients with and without ventilation or between patients with and without long-term hospitalization.

The correlation of sCLEC-2 levels with platelet count was poor in the present study. While the plasma sCLEC-2 levels are usually considered to be correlated with platelet count in patients without platelet consumption [14], marked consumption of platelets is observed in DIC and Pre-DIC [20].

## 5. Conclusions

The plasma levels of sCLEC-2 were significantly higher in patients with DIC or Pre-DIC and related with a poor outcome.

## Figures and Tables

**Figure 1 jcm-10-02860-f001:**
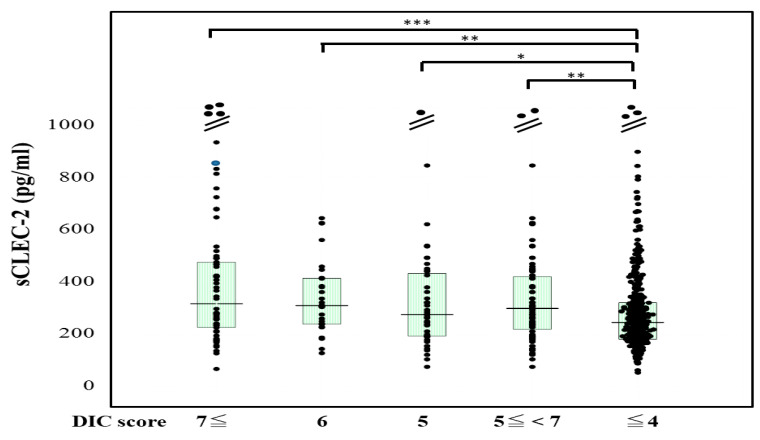
sCLEC-2 levels in patients with DIC, those with pre-DIC and those without DIC. DIC, DIC score ≥ 7; pre-DIC, DIC score 5 or 6; without DIC, DIC score ≤ 4; * *p* < 0.05; ** *p*< 0.01; *** *p* < 0.001.

**Figure 2 jcm-10-02860-f002:**
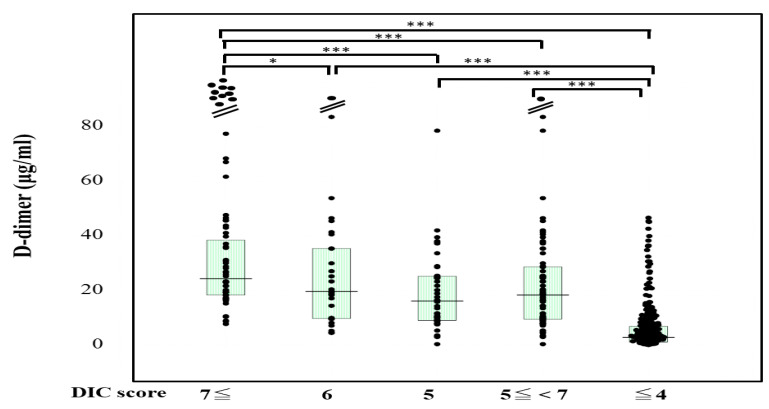
D-dimer levels in patients with DIC, those with pre-DIC and those without DIC. DIC, DIC score ≥ 7; pre-DIC, DIC score 5 or 6; without DIC, DIC score ≤ 4; * *p* < 0.05; *** *p* < 0.001.

**Figure 3 jcm-10-02860-f003:**
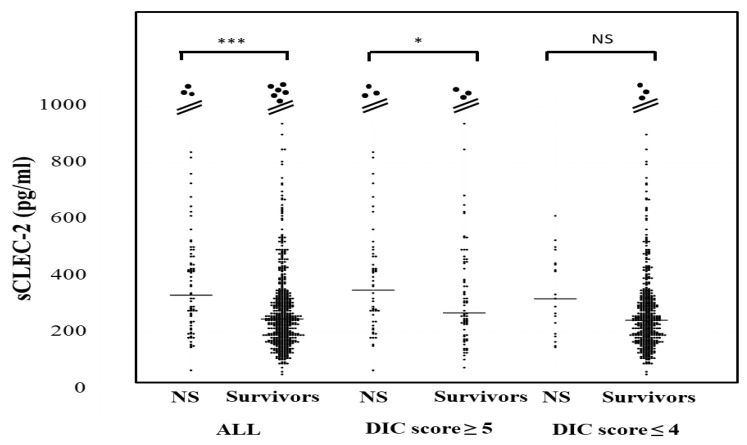
sCLEC-2 levels in survivors and non-survivors. NS; non-survivor; * *p* < 0.05; *** *p* < 0.001; NS. not significant.

**Figure 4 jcm-10-02860-f004:**
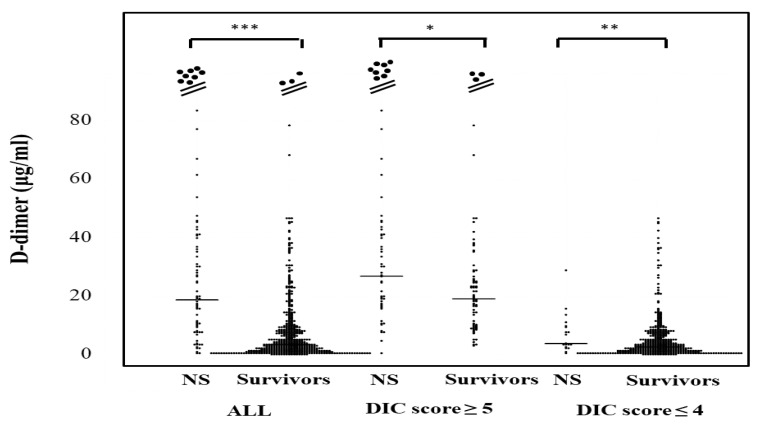
D-dimer levels in survivors and non-survivors. NS; non-survivor; * *p* < 0.05; ** *p*< 0.01; *** *p* < 0.001; NS. not significant.

**Table 1 jcm-10-02860-t001:** Subjects.

Disease	Age (Years)	Sex (F:M)	PT–INR	PLT (10^9^/L)	sCLEC–2 (pg/mL)	D–Dimer (μg/mL)	CRP (mg/dl)	DIC Score ≥ 5	Mortality
Infection	76.5	109:137	1.13 ***	2.07 **	379 ***	4.7 ***	9.2 ***	60	25
	(57.0–83.0)		(1.03–1.25)	(1.33–2.64)	(275–798)	(1.8–11.1)	(3.0–16.5)	(24.4%)	(10.2%)
Aortic aneurysm	72.0	30:45	1.06 ***	1.82 ***	307 ***	7.4 ***	1.9 ***	9	3
	(50.0–78.0)		(0.99–1.18)	(1.42–2.17)	(246–489)	(3.2–13.2)	(0.2–7.3)	(12.0%)	(4.0%)
Trauma#	69.0	32:30	1.02 **	2.18	247 ***	6.2 ***	0.6 ***	13	5
	(45.0–80.0)		(0.93–1.08)	(14.1–27.2)	(177–353)	(2.5–22.2)	(0.1–3.0)	(21.0%)	(8.1%)
CPA	79.5	9:23	1.48 ***	1.16 **	652 ***	23.6 ***	0.6 ***	25	29
	(66.0–86.5)		(1.13–1.94)	(0.71–2.26)	(440–1162)	(9.0–44.7)	(0.2–4.8)	(78.1%)	(90.6%)
Solid cancer	72.5	14:18	1.06 ***	2.43	279 ***	4.1 ***	2.9 ***	7	8
	(67.5–79.5)		(0.99–1.18)	(1.70–3.14)	(194–365)	(2.5–15.1)	(0.7–9.9)	(21.9%)	(25.0%)
Obstetric disease	30.0	37:0	0.98	2.32	207	2.5 ***	0.2	3	0
	(30.0–42.0)		(0.94–1.04)	(1.63–2.52)	(153–276)	(1.7–7.8)	(0.1–0.3)	(8.1%)	(0%)
Other DIC/ Pre–DIC	78.0	4:7	1.51 ***	1.32 ***	424 ***	21.6 ***	9.4 ***	11	4
	(66.3–84.3)		(1.31–1.70)	(0.72–1.32)	(257–464)	(19.2–39.2)	(4.0–27.2)	(100%)	(36.4%)
UCS	61.0	43:37	0.96	2.34	194	0.7	0.1	0	0
	(48.0–73.5)		(0.92–1.00)	(1.87–2.75)	(146–250)	(0.4–1.6)	(0.0–0.3)	(0%)	(0%)

Data are presented as the median (25th–75th percentile). *** *p* < 0.001 or ** *p* < 0.05 in comparison with UCS. Trauma#, trauma (including burn and hypothermia); CPA, cardiopulmonary arrest; Other DIC/Pre-DIC; DIC or Pre-DIC due to other underlying diseases; UCS, unidentified clinical syndrome; PT, prothrombin time; PLT, platelet count.

## Data Availability

The data presented in this study are available on request to the corresponding author. The data are not publicly available due to privacy restrictions.

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
