# Peer review of "Soluble C-Type Lectin-Like Receptor 2 Is a Biomarker for Disseminated Intravascular Coagulation"

_jcm, 2021, doi:10.3390/jcm10132860_

Round 1

Reviewer 1 Report

The authors addressed an important gap of knowledge trying to characterize DIC and pre-DIC conditions, that is of undoubtful clinical utility. Nevertheless, the proposed study presents some major issues that compromise its scientific soundness.

  1. sCLEC-2 is expressed on platelets AND a number of immune cells. Thus, sCLEC-2 levels are expected to rise in sepsis and inflammation, conditions that frequently lead to DIC. Additional biomarkers should be explored in this series of patients in order to rule out whether sCLEC-2 is specifically associated with platelet activation or it is induced by the contemporary inflammation or sepsis (in particular, in the patients with cardiac conditions where elevation of sCLEC-2 has been described in the absence of DIC). In case that this is not possible, the overall message should be downgraded to represent this limitation.
  2. It is not clear from the Methods section the timing of the blood sampling from the patients. In particular, it is recognized that DIC is composed of two distinct phases - the first one characterized by an overactivation of the coagulation system with a hypercoagulable status and parameters, followed by the second phase that is an (irreversible) exhaustion of the coagulation system. The diagnosis of DIC may be done at any time. Patient management leading to enrollment (inclusion and exclusion criteria) deserves a more detailed description.
  3. A statistical analysis of sCLEC-2 levels and outcome (other than death vs. survival, for instance - mechanical ventilation duration, requirement for transfusion of blood products, length of hospital stay, etc.) may add value to the study. 
  4. Discussion - a more thorough discussion of the limitations of the study is needed. 

Minor issues:

  1. Please consider to change the unit of measure of platelet count to more frequent 10x9/L or 10x3/uL.

Author Response

We thank you for your helpful and valuable comments on our study titled, “Soluble C-type lectin-like receptor 2 is a biomarker for disseminated intravascular coagulation.” (Ref.: Manuscript Number: jcm-1246451). We have responded to all suggestions from the reviewers and have revised the manuscript accordingly. Our corrections are indicated as red letters with yellow highlight in the revised manuscript.

Respectfully yours,

Hideo Wada

Responses to the reviewer’s comments

The authors addressed an important gap of knowledge trying to characterize DIC and pre-DIC conditions, that is of undoubtful clinical utility. Nevertheless, the proposed study presents some major issues that compromise its scientific soundness.

Comment 1

sCLEC-2 is expressed on platelets AND a number of immune cells. Thus, sCLEC-2 levels are expected to rise in sepsis and inflammation, conditions that frequently lead to DIC. Additional biomarkers should be explored in this series of patients in order to rule out whether sCLEC-2 is specifically associated with platelet activation or it is induced by the contemporary inflammation or sepsis (in particular, in the patients with cardiac conditions where elevation of sCLEC-2 has been described in the absence of DIC). In case that this is not possible, the overall message should be downgraded to represent this limitation.

Response 1CRP was added in Table 1 as additional biomarker for inflammation. The patients with cardiac conditions may include the patients with cardiopulmonary arrest (CPA). Although CPA is not considered an appropriate for underlying disease of DIC, platelet activation can occur in CPA and is associated with DIC.

Comment 2

It is not clear from the Methods section the timing of the blood sampling from the patients. In particular, it is recognized that DIC is composed of two distinct phases - the first one characterized by an overactivation of the coagulation system with a hypercoagulable status and parameters, followed by the second phase that is an (irreversible) exhaustion of the coagulation system. The diagnosis of DIC may be done at any time. Patient management leading to enrollment (inclusion and exclusion criteria) deserves a more detailed description.

Response 2The timing of sampling and patient management are described in detail. The inclusion and exclusion criteria were described.

Comment 3

A statistical analysis of sCLEC-2 levels and outcome (other than death vs. survival, for instance - mechanical ventilation duration, requirement for transfusion of blood products, length of hospital stay, etc.) may add value to the study. 

Response 3There were no significant differences in the plasma sCLEC-2 levels between the DIC and pre-DIC patients with ventilation (310pg/ml; 240-468 pg/ml) and patients without ventilation (293 pg/ml; 193-468 pg/ml) or between patients with ≤28 days hospitalization (321 pg/ml; 219-468 pg/ml) and those with ≥29 days hospitalization (304 pg/ml; 227-475 pg/ml). Only 9 patients with DIC or Pre-DIC were treated with blood transfusion.

Comment 4

Discussion - a more thorough discussion of the limitations of the study is needed. 

【Response 4】The limitations of the study are discussed.

Comment 5

Minor issues:

Please consider to change the unit of measure of platelet count to more frequent 10x9/L or 10x3/uL.

Response 5The platelet count was expressed as x109/L

Reviewer 2 Report

This is an excellent manuscript. The study provides important novel results to advance the field of DIC. The results of this thoroughly compiled manuscript demonstrate important tools to improve the management of patients with DIC and Pre-DIC. Therefore, it is very deserving to investigate plasma sCLEC-2 as a new biomarker for DIC/Pre-DIC. It also seems to be an important biomarker for prognostic outcome of DIC.

Author Response

We thank you for your helpful and valuable comments on our study titled, “Soluble C-type lectin-like receptor 2 is a biomarker for disseminated intravascular coagulation.” (Ref.: Manuscript Number: jcm-1246451). We have responded to all suggestions from the reviewers and have revised the manuscript accordingly. Our corrections are indicated as red letters with yellow highlight in the revised manuscript.

Respectfully yours,

Hideo Wada

Responses to the reviewer’s comments

Response 6Thank you very much. I carefully revised this manuscript in accordance with comments from reviewer 1.

Round 2

Reviewer 1 Report

My comments were thoroughly addressed and I have no further comments for the authors.